# Identification and Validation of UPF1 as a Novel Prognostic Biomarker in Renal Clear Cell Carcinoma

**DOI:** 10.3390/genes13112166

**Published:** 2022-11-20

**Authors:** Chun Wu, Hongmu Li, Wuguang Chang, Leqi Zhong, Lin Zhang, Zhesheng Wen, Shijuan Mai

**Affiliations:** 1State Key Laboratory of Oncology in South China, Collaborative Innovation Center for Cancer Medicine, Sun Yat-sen University Cancer Center, Guangzhou 510060, China; 2Department of Thoracic Oncology, Sun Yat-sen University Cancer Center, Guangzhou 510060, China

**Keywords:** UPF1, clear cell renal cell carcinoma, immune microenvironment, bioinformatics, pan-cancer

## Abstract

**Background:** Up frameshift protein 1 (UPF1) is a key component of nonsense-mediated mRNA decay (NMD) of mRNA containing premature termination codons (PTCs). The dysregulation of UPF1 has been reported in various cancers. However, the expression profile of UPF1 and its clinical significance in clear cell renal cell carcinoma (ccRCC) remains unclear. **Methods:** In order to detect UPF1 expression in ccRCC and its relationship with the clinical features of ccRCC, bulk RNA sequencing data were analyzed from The Cancer Genome Atlas (TCGA), Gene Expression Omnibus (GEO) and ArrayExpress databases. The impact of UPF1 on the immune microenvironment of ccRCC was evaluated by multiple immune scoring algorithms to identify the cell groups that typically express UPF1 using ccRCC single cell sequencing (scRNA) data. In addition, genes co-expressed with UPF1 were identified by the weighted gene correlation network analysis (WGCNA), followed by KEGG and Reactome enrichment analysis. A series of functional experiments were performed to assess the roles of UPF1 in renal cancer cells. Finally, pan-cancer analysis of UPF1 was also performed. **Results:** Compared with normal tissues, the expression levels of UPF1 mRNA and protein in tumor tissues of ccRCC patients decreased significantly. In addition, patients with low expression of UPF1 had a worse prognosis. Analysis of the immune microenvironment indicated that UPF1 immune cell infiltration was closely related and the ccRCC scRNA-seq data identified that UPF1 was mainly expressed in macrophages. WGCNA analysis suggested that the functions of co-expressed genes are mainly enriched in cell proliferation and cellular processes. Experimental tests showed that knockdown of UPF1 can promote the invasion, migration and proliferation of ccRCC cells. Lastly, pan-cancer analysis revealed that UPF1 disorders were closely associated with various cancer outcomes. **Conclusions:** UPF1 may play a tumor suppressive role in ccRCC and modulate the immune microenvironment. The loss of UPF1 can predict the prognosis of ccRCC, making it a promising biomarker and providing a new reference for prevention and treatment.

## 1. Introduction

Renal cell carcinoma (RCC) accounts for about 2% of all malignant tumors in the world and is the tumor with the highest mortality rate of urogenital tumors [1,2]. The most common type of RCC is the clear cell renal cell carcinoma (ccRCC), accounting for around 75% of all renal malignancies [3]. Surgical resection is the most effective strategy for the treatment of ccRCC. Unfortunately, distant metastases are already present in 25% to 30% of newly diagnosed ccRCC patients [4]. Therefore, exploring the potential molecular mechanism of ccRCC progress and developing effective diagnostic and prognostic biomarkers are crucial for accurate early diagnosis, improved prognosis, and rational individualized treatment strategies.

Recently, the role of up-frameshift protein 1 (UPF1) in tumorigenesis has been widely investigated. As a central protein required for the nonsense-mediated mRNA degradation pathway (NMD), UPF1 selectively recognizes and degrades mRNAs with premature termination codon (PTC)-containing transcripts (PTCs) via a complex set of NMD factors to halt translation, thus protecting cells from aberrant toxic transcripts [5]. Several studies have indicated that UPF1 is downregulated and related to poor prognosis in pancreatic adenosquamous carcinoma (PASC) [6], hepatocellular carcinoma (HCC) [7], gastric cancer (GC) [8], inflammatory myofibroblastic tumors (IMT) [9], thyroid cancer (TC) [10], ovarian cancer (OC) [11] and glioma [12]. Controversially, UPF1 was also found highly expressed in glioblastoma and lung adenocarcinoma [13,14]. Notably, the relationship between UPF1 and renal cancer has not been discussed so far.

In this study, we identified that UPF1 has low expression in ccRCC and its lower expression level can predict poorer prognosis of ccRCC patients according to a set of bioinformatic analyses. Further, cellular experiments proved that knockdown of UPF1 enhanced the invasion, migration, and proliferation abilities in renal cancer cell lines. Western blot assay showed that the phosphorylated AKT protein increased after UPF1 knockdown. Additionally, a close relationship between UPF1 expression level and tumor immune microenvironment was explored by analyzing the GSE data of ccRCC single cell sequencing (scRNA). Finally, the co-expressed genes of UPF1 and their roles in oncogenesis were also identified through a pan-cancer analysis. The framework of our study is illustrated in Figure 1.

## 2. Materials and Methods

### 2.1. Data Acquisition

The transcriptomic and clinical data of ccRCC patients were obtained from The Cancer Genome Atlas (TCGA-KIRC, www.cancer.gov/tcga, accessed on 10 August 2022). The microarray expression data (GSE53757 [15] and GSE40435 [16]) and scRNA-seq data (GSE159115 [17]) of ccRCC were obtained from the Gene Expression Omnibus (GEO, www.ncbi.nlm.nih.gov/geo/, accessed on 10 August 2022) database, and gene expression and clinical data of external ccRCC cohort were acquired from ArrayExpress: MTAB-1980 (https://www.ebi.ac.uk/arrayexpress, accessed on 10 August 2022).

### 2.2. Single-Cell RNA Sequencing Data Preprocessing

Single-cell RNA sequencing (scRNA seq) data were processed by the R package “Seurat” [18]. Seurat objects were created for each sample with using the ‘CreateSeuratObject’ function. Cells with a high mitochondrial gene percentage of >15%, and gene number detection <300 or >5000 were considered low-quality cells and discarded. Then, the “NormalizeData” package was employed to normalize the data, and the ‘FindVariableFeatures’ function was performed to identify the highly variable genes. Subsequently, we determined the different cell types with default parameters, and cells were clustered using the ‘FindClusters’ function into 26 different cell clusters, which were then visualized using UMAP. We identify differentially expressed genes (DEGs) in each cluster via the ‘FindAllMarkers’ function. Finally, a few classical markers of cell subset definition were obtained from previous studies [19,20] and manually annotated according to marker expression.

### 2.3. UPF1 Expression Analysis and Survival Analysis

The expression values were processed and normalized using the published protocols [21]. The expression levels of mRNA UPF1 were examined in TCGA and GEO cohorts, and divided into two groups based on the median expression: the high UPF1 expression group and the low UPF1 expression group. Then, the Kaplan-Meier survival curves were analyzed by the R package “survival”. The differentially expressed genes (DEGs) between the high UPF1 expression and low UPF1 expression group were screened out using the R package “DESeq2”. Furthermore, an online tool called Metascape [22] (https://metascape.org/, accessed on 15 August 2022) was utilized to perform functional enrichment analysis on upregulated DEGs in the high UPF1 group and DEGs upregulated in low UPF1 groups. The data of mutations were downloaded and visualized through the “maftools” package [23].

### 2.4. Immunological Features of the TME in ccRCC

To explore the relationship between UPF1 and the tumor microenvironment (TME), we systematically analyzed the immunomodulators, inhibitory immune checkpoints, and immune cells in ccRCC samples. Briefly, we summarized and analyzed 122 immunomodulators (chemokines, immune stimulators, MHCs, and receptors) [24]. Then, we used seven distinct algorithms (ssGSEA, EPIC, MCP-counter, xCell, quanTIseq, CIBERSORTx, and TIMER) [25,26,27,28,29,30,31] to determine the level of tumor-infiltrating immune cells (TIICs) infiltration in the TME. Furthermore, the immune-related pathway from the Immport database [32] (http://www.immport.org, accessed on 15 August 2022) and the Reactome [33] pathways activities between the high UPF1 expression group and low UPF1 expression group were compared by the R package “GSVA” [34].

### 2.5. Weighted Gene Co-Expression Network Analysis (WGCNA)

In order to explore the correlation between gene modules and UPF1 expression status in the TCGA and GEO cohorts [35], the R package “WGCNA” was used to construct gene co-expression networks. The independent GEO datasets GSE53757 and GSE40435 were merged utilizing the R package “inSilicoMerging” [36], and then the “sva” R package was used to perform batch correction on the merged data to adjust batch effects in the microarray expression data [37], and the merged microarray was employed for WGCNA. A power of β = 5 (scale-free R^2^ = 0.88) in the GEO cohort and a power of β = 16 (scale-free R^2^ = 0.85) were selected as soft-threshold parameters to ensure unsigned scale-free co-expression gene networks. The module with the highest correlation coefficient and the most significant *p* value was determined as the key module, and genes in the key module were selected for further analysis. Then, the online tool Metascape was employed to perform KEGG and Reactome functional enrichment analysis between key modules in the TCGA and GEO cohort. Lastly, by overlapping the key module genes in the TCGA and GEO cohorts, UPF1-related co-expression genes were identified and their expression pattern and GO functions were explored using the R package “clusterProfiler” [38]. 

### 2.6. Analysis of UPF1 in Pan-Cancer

TCGA provided expression data and corresponding clinical information for 31 types of cancer. The expression status of UPF1 was analyzed and visualized by GEPIA2 [39] (http://gepia2.cancer-pku.cn/, accessed on 15 August 2022). Then, we investigated the correlation between UPF1 expression levels and patients’ survival for each cancer type [40].

### 2.7. Patients and Samples

Ten cases of snap-frozen ccRCC tissues and their corresponding adjacent non-tumor specimens were collected from the Tumor Resource Bank of Sun Yat-sen University Cancer Center (SYSUCC) and stored at −80 °C before qRT-PCR assay. Additionally, 10 formalin-fixed, paraffin-embedded primary ccRCC specimens obtained from the Department of Pathology in SYSUCC were used for the immunohistochemistry (IHC) assay. There was no radiotherapy or chemotherapy before biopsy sampling in any of the patients. This study was performed with the approval of the Ethics Committee of Sun Yat-sen University Cancer Center (GZR2022-346).

### 2.8. qRT-PCR and IHC

Total RNA was extracted from 10 sets of ccRCC tissues and their adjacent non-cancer tissues by TRIzol reagent (TIANGEN, Beijing, China). The concentration and purity of RNA was determined with the help of UV spectrophotometer. The cDNA was amplified by reverse transcriptase. Each sample was measured in triplicate. The following PCR primers were used for amplification: UPF1, 5′-CCTTCCCATCCAACATCTTC-3′ (forward), 5′-AACATCGGTTTATCGGGTTG-3′ (reverse); GAPDH as an endogenous control, 5′-ATCAAGAAGGTGGTGAAGCAGG-3′ (forward), 5′-CGTCAAAGGTGGAGGAGTGG-3′ (reverse). The 2^−ΔΔCT^ method were used to calculated UPF1’s relative expression levels. A rabbit anti-UPF1 antibody (1:200, Abcam, Cambridge, MA, USA) was used in IHC analyses of UPF1. Paraffin sections were deparaffinized and hydrated in water with 3% H_2_O_2_ blocked endogenous peroxidase. After microwave repaired, the first antibody was incubated at 37 °C for 1 h. The second antibody labeled with peroxidase was incubated at 37 °C for 45 min. Then, the colors were created with chromogenic solutions DAB [41]. 

### 2.9. Cell Culture and Transfection

The renal cancer cell lines ACHN and Caki-1 were obtained from Xinyuan Biotech Co. Ltd. (Shanghai, China) and authenticated using short tandem repeat profiling. Cells were cultured in RPMI-1640 (Gibco, Grand Island, NY, USA) containing 10% fetal bovine serum (Sigma, Shanghai, China) at 37 °C and 5% CO_2_. LV3 vectors were used to clone shRNA1 targeting UPF1. The following shRNA sequences were used: UPF1 shRNA, 5′-CCUACCAGUACCAGAACAUTT-3′. Lipofectamine 3000 (Invitrogen, CA, USA) was used for transfection according to the manufacturer’s guidelines.

### 2.10. Western Blotting

The RIPA lysis buffer (Roche, Basel, Switzerland) was used to lyse cellular proteins, and the protein concentrations were determined using BCA Protein Assay Kit (Beyotime, Shanghai, China). The primary antibodies were incubated overnight at 4 °C with UPF1 (1:1000, Abcam), AKT (1:1000, CST, MA, USA) and P-AKT (1:1000, CST). Before exposure imaging, goat anti-rabbit secondary antibodies (1:5000, Proteintech, Wuhan, China) were incubated at room temperature for 1 h.

### 2.11. Cell Counting Kit-8 Assay

A CCK-8 assay (JingXin Biological Technology, Guangzhou, China) was used to measure cell proliferation. Cells from ACHN, Caki-1, and HK2 were prepared into cell suspensions with a density of 5 × 10^3^ cells/mL. There were 96-well plates containing 1 × 10^3^ cells/well, each cultured in 5% CO_2_ in a 37 °C incubator. During the second hour of the measurement, ten microliters of CCK-8 solution was added to culture and repeated for a total of 24, 48, 72, 96, 120, and 144 h. Microplate readers were used to measure absorbance at 450 nm after incubation.

### 2.12. Migration and Invasion Assays

For in vitro migration assays, cells (4 × 10^4^) in 200µL of serum-free medium were seeded in the upper chambers of the Transwell plates (Corning, NY, USA) with 8 um pores. Lower chambers were chemoattracted with FBS. After 24 h of incubation, the cells in the upper chamber were removed. To perform in vitro invasion assays, cells (8 × 10^4^) were seeded in a Matrigel-coated chamber (BD Biosciences, CA, USA) with 8 mm pores present in the insert of a 24-well culture plate (BD Biosciences) and lower chambers were chemoattracted with FBS. Upper chamber cells were removed 48 h after incubation. Migrative and invasive cells on the lower side of the chamber were fixed with 100% methanol for 10 min and stained with crystal violet for 30 min at room temperature. Cells were counted under a microscope in four random fields per well. Experiments were performed three times.

### 2.13. Statistics

The Kaplan–Meier method was used to assess OS or FPS, and a two-sided *p*-value of less than 0.05 was regarded as significant. All correlation analyses were conducted using Spearman correlation analysis. Statistics were performed using GraphPad Prism 9.0, with statistical significance defined as a *p*-value less than 0.05. Depending on the experiment, t-tests were either paired or unpaired.

## 3. Results

### 3.1. UPF1 Has Low Expression in ccRCCs and Was Correlated with Poor Prognosis

To explore the expression and clinical correlation of UPF1 in ccRCCs, a range of bioinformatics databases were analyzed. We found that the UPF1 expression level was significantly lower in ccRCC tissue samples compared with normal renal tissues at the mRNA level (Figure 2A–D). Further analysis from TCGA databases was performed via an alluvial diagram (Appendix A), which revealed that the mRNA level of UPF1 was relatively lower in advanced and high-grade ccRCC tissues (Figure 2E–H). In addition, ccRCC patients with lower UPF1 expression had a poorer overall survival (OS) (Figure 2E). A similar relationship was shown between UPF1 expression and progression-free survival (PFS) and disease-specific survival (DSS) of patients with ccRCC based on TCGA data (Figure 2F,G). In an external independent cohort (MTAB-1980), a similar trend was also observed (Figure 2H). 

### 3.2. The Effects of UPF1 in the ccRCC Tumor Microenvironment 

Recently, UPF1 has been found to be closely related to the tumor immune microenvironment [42]. We next examined the relationship between the UPF1 and immune-related characteristics in ccRCC, which indicated that UPF1 was positively correlated with a majority of immunomodulators, including chemokines, immunostimulators, MHC and receptors (Figure 3A). A positive relationship was also found between UPF1 and immune checkpoint inhibitors, including CD274, PVR, CEACAM1, CD276 and CD200 (Figure 3B). 

To further investigate the relationship between UPF1 and immunocyte infiltration, the ssGSEA, EPIC, MCP-counter, xCell, quanTIseq, CIBERSORTx and TIMER algorithms were employed to compute the levels of diverse types of immunocytes base on TCGA-KIRC data. A total of 148 cell type indexes were evaluated, and significant correlations were observed between UPF1 and most immunocytes. Interestingly, more T cells and macrophage infiltration was found in the high-UPF1 group (Figure 3C). Distinct pathway activity was identified by “GSVA”. For pathways from the Immport database, cytokine_receptors, TGF-β_family_member_receptor and TCR signaling pathways were activated in the high-UPF1 group, while interferons, interleukins, interferon_receptor, antimicrobials and chemokines-related pathways were activated in the low-UPF1 group (Figure 4A). Reactome data showed higher pathway activity in the high-UPF1 group than in the low-UPF1 group for pathways from the database (Figure 4B).

In order to specify the immune cell types expressing UPF1, we analyzed the ccRCC scRNA sequencing data. After the data were normalized (Appendix A), the cells were clustered into immune cells and non-immune cells (Figure 4C,D). Immune cells were divided into five sub-clusters, in which macrophages were the major cell type expressing UPF1 (Figure 4E,F).

### 3.3. Screen Key Modules and Co-Expression Genes of UPF1

Gene co-expression networks were constructed based on TCGA and GEO cohorts to identify genes associated with UPF1 in ccRCC patients (Appendix A). For the merged GEO data, batch effects were corrected (Appendix A). A total of 30 modules were identified in the GEO cohort, and 16 modules were identified in the TCGA cohort. Among them, we found that the light-yellow module in the TCGA cohort and yellow module in the GEO cohort were most statistically correlated with UPF1 groups (Figure 5A). Then, the genes in these two key modules were subjected to the Metascape tool for KEGG and Reactome enrichment analysis, and we found that these modules were both enriched in pathways involving cell proliferation and cellular processes, i.e., signaling by WNT and endocytosis pathways (Figure 5B). 

Subsequently, by taking the intersection of genes in the two key modules, ten UPF1-related co-expression genes (*MMS19*, *CABIN1*, *GOLGA2*, *ASXL1*, *TNPO2*, *BSDC1*, *AP2A2*, *TRIM56*, *DCTN1*, and *ACTN4*) were identified (Figure 6A), and the expression levels of these genes were found to be significantly positively correlated with UPF1 (Figure 6B). Lastly, we identified 20 GO terms that these genes were involved in, which include microtubule nucleation and the peroxisome proliferator activated receptor signaling pathway (Figure 6C).

### 3.4. Experiments to Validate the Expression and Functions of UPF1

The qRT-PCR results from 10 paired ccRCC samples collected from our Cancer Center showed that UPF1 expression was significantly lower in cancer tissues than in normal kidney tissues (*p* < 0.05) (Appendix A, Figure 7A). These results were consistent with those obtained from bioinformatics analysis. IHC in paraffin-embedded tumor samples from 10 ccRCC patients also showed that UPF1 protein staining was relatively weaker in the tumor cells compared with that in the adjacent non-tumor kidney tissues (Appendix A, Figure 7B). Furthermore, CCK8 assay indicated that UPF1 knockdown significantly enhanced the proliferation of ACHN and Caki-1 cell lines (Figure 7C,D). Moreover, after knockdown of UPF1, the migration and invasion capacities of the ACHN and Caki-1 cells were also dramatically enhanced (Figure 7E,F). We also knocked out UPF1 in the normal renal epithelial cell lines HK2, and the CCK8 increment curve suggested that the knockdown of UPF1 would also promote the proliferation of the normal renal cell line (Appendix A). In summary, our results showed that UPF1 might inhibit the proliferation, migration and invasion abilities of renal cancer cells. PTEN regulation was detected in the enrichment analysis of UPF1 co-expression genes. Since PTEN is the key upstream regulator of the AKT signalling pathway, we conducted a Western blot assay and found that the phosphorylated form of AKT obviously increased, suggesting the activation of the AKT signal pathway after UPF1 knockdown in ACHN and Caki-1 cells (Figure 7G,H).

### 3.5. Analysis of UPF1 in Pan-Cancer

The GEPIA2 analysis results suggested that UPF1 show low expression in various tumors including Kidney Clear Cell Carcinoma (KIRC), Lung Adenocarcinoma (LUAD), Ovarian Cancer (OV), Melanoma (SKCM) and Testicular Cancer (TGCT) (Figure 8A). Pan-cancer cohorts in the TCGA datasets were divided into the high UPF1 group and low UPF1 group, according to the median value of UPF1 expression. The results showed that compared with high expression of UPF1, low expression of UPF1 was associated with a poor prognosis in most cancers, including disease specific survival (DSS), overall survival (OS) and progression free interval (PFI) (Figure 8B).

## 4. Discussion

NMD is an evolutionarily conserved surveillance mechanism of post-transcriptional gene regulation in both normal and pathological processes. Aberrant transcripts containing PTCs and mRNAs with extended 3′-UTRs are targeted by NMD for rapid elimination to prevent the synthesis of truncated proteins that could be detrimental to the cell [43]. UPF1, an ATP-binding RNA helicase, is the core factor required for NMD in all eukaryotes. A growing body of evidence shows that UPF1 is dysregulated in multiple tumors and promotes cell survival during tumorigenesis [44]. UPF1 can act as a tumor suppressor to induce apoptosis of tumor cells [45], inhibit cell proliferation [46] and weaken the cell stemness of tumor [47]. On the contrary, UPF1 may also play an oncogenic role in cancer by promoting proliferation, migration, invasion, apoptosis and colony-forming ability, as well as have CSC-like characteristics [8]. However, the expression profile of UPF1 and its role in ccRCC still remain unclear.

Our study found that UPF1 expression levels were significantly lower in ccRCC tissues when compared with adjacent non-tumor tissues. Moreover, lower UPF1 expression was significantly correlated with more advanced clinical stage and poorer differentiation state in ccRCC cases. Importantly, low UPF1 expression was significantly related to worse OS, PFS and DSS of ccRCC patients, based on multiple database analysis. To confirm the expression level of UPF1, we collected paired ccRCC samples from our Cancer Center and performed qRT-PCR and IHC assays, and the results were consistent with the bioinformatic analyses. 

In order to investigate the potential mechanism of UPF1 in ccRCC, the GSVA algorithm was used to screen out the most important signaling pathways by comparing Immport and Reactome pathway activities between the low UPF1 group and the high UPF1 group, which indicated that the high-UPF1 group was enriched in cytokine receptor interaction. This may suggest that UPF1 plays its anti-cancerous role through increasing the number of cytokine receptors, as previously reported [48,49,50]. 

Next, seven independent proven algorithms were utilized to evaluate the immunocyte proportion in each sample to comprehend the landscape of immunocyte infiltration status of the ccRCC tumor microenvironment, and T cells and macrophages were identified as significantly correlated with UPF1. Consistently, results from the single-cell sequencing data analysis also indicated that UPF1 was mainly expressed in the macrophages of ccRCC patients. As previously reported, macrophages play a substantial role in immunomodulation [51], and a UPF1-deficiency has been reported to cause immature T cell development [52]. All in all, UPF1 is closely related to immune cells, and further research will help find new therapeutic targets for ccRCC. In addition, key co-expression genes related to UPF1 were screened out by building co-expression networks, and functional enrichment analysis implicated that they were mainly involved in the cell proliferation and cellular processes, including the WNT and endocytosis pathways.

CcRCC cells were examined for the role of endogenous UPF1. UPF1 was silenced in ACHN and Caki-1 cells by using specific siRNA against UPF1. A CCK8 incorporation assay and Transwell assays confirmed that the downregulation of UPF1 promoted the proliferation, migration and invasion of ccRCC cells. Knockdown of UPF1 in normal renal cell line can also promote proliferation. PTEN regulation was detected in the enrichment analysis of UPF1 co-expression genes. Since PTEN is the key upstream regulator of the AKT signaling pathway, we conducted a Western blotting and detected that the AKT signaling might be activated after UPF1 is knocked out. However, further experiments need to be conducted to unveil the molecular mechanisms of UPF1 in ccRCC.

Finally, pan-cancer analysis showed that poor prognosis was associated with low expression of UPF1. In previous studies, UPF1 was found to be significantly down regulated in pancreatic cancer [6], hepatocellular carcinoma (HCC) [46], gastric cancer (GC) [8], thyroid cancer (TC) [10] and glioma [12], and was verified in most cell lines of these cancers. However, Ha and Bokhari A found that UPF1 was significantly upregulated in LADC [14] and CRC (in MSI) [53]. We propose that there might be two main reasons for the inconsistent expression trend of UPF1 in different cancers. Firstly, it was reported that oxidative stress and endoplasmic reticulum (ER) stress could suppress NMD through the phospho-eIF2α/ATF4 pathway [54]. This might explain the low expression of UPF1 in renal cancer, which has feature enhanced oxidative stress. Secondly, tumors with high tumor mutation burdens (TMB) tend to generate a high number of PTC-mRNAs, which will trigger NMD and account for increased expression levels of NMD factors [53]. These findings indicate that the expression of UPF1 is inconsistent in different cancers. However, the mechanism of UPF1 in these cancers needs further study.

## 5. Conclusions

UPF1 may play a tumor suppressive role in ccRCC and is involved in the regulation of the immune microenvironment. The loss of UPF1 can promote the progress of ccRCC, which can be used as a promising biomarker of ccRCC and provide a new reference for prevention and treatment. 

## Figures and Tables

**Figure 1 genes-13-02166-f001:**
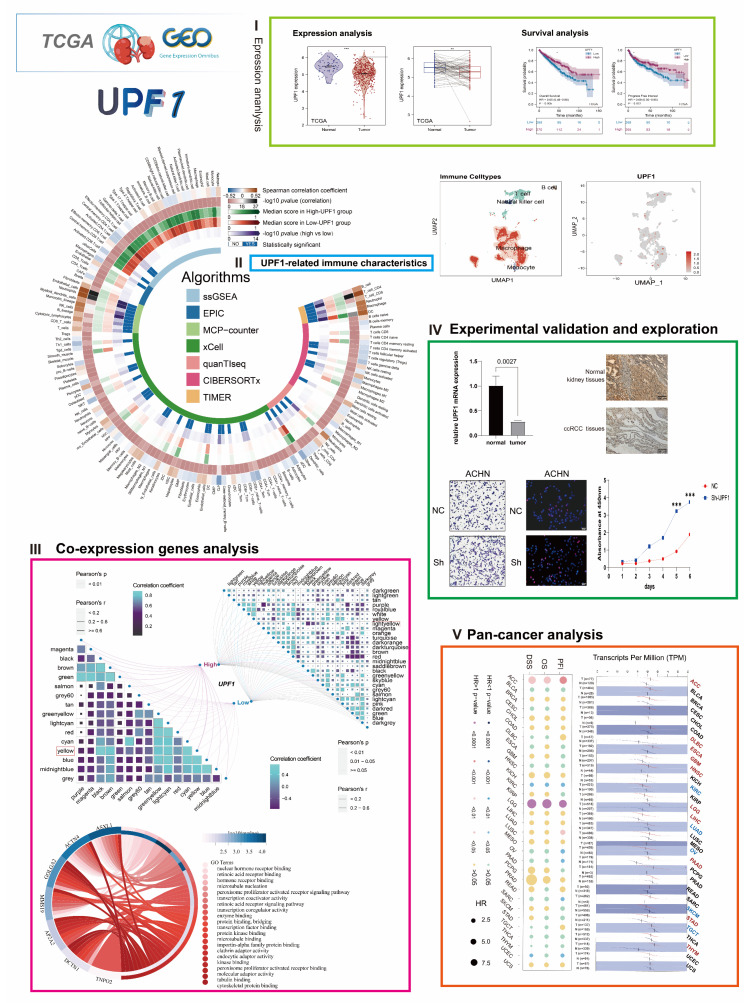
The flow-process diagram of the current study. (**I**) Expression analysis. (**II**) UPF1-related immune characteristics. (**III**) Co-expression genes analysis. (**IV**) Experimental validation and exploration. (**V**) Pan-cancer analysis of UPF1. ** *p* < 0.01 and *** *p* < 0.001.

**Figure 2 genes-13-02166-f002:**
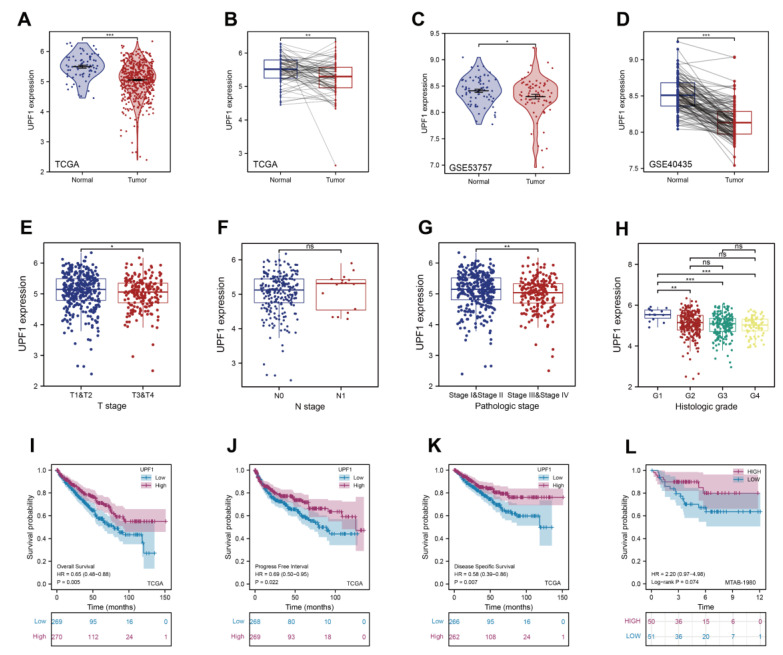
Characteristics of UPF1 expression status in ccRCC tissue samples from databases. (**A**) The expression of UPF1 based on TCGA data. (**B**) TCGA data (paired samples) showing UPF1 expression. (**C**) GEO-based expression of UPF1 (GSE53757). (**D**) GEO-based expression of UPF1 (GSE40435). (**E**) Relationship between UPF1 expression and T stage. (**F**) The relationship between UPF1 expression and Lymph node metastasis. (**G**) The relationship between UPF1 expression and Pathological stage. (**H**) The relationship between UPF1 expression and Histologic grade. The effect of UPF1 expression on ccRCC patients’ overall survival (**I**), progression-free survival (**J**) and disease-specific survival (**K**) based on the TCGA. (**L**) The effect of UPF1 expression on ccRCC patients’ overall survival based on the MTAB-1980 cohort. * *p* < 0.05, ** *p* < 0.01 and *** *p* < 0.001. ns: no significance.

**Figure 3 genes-13-02166-f003:**
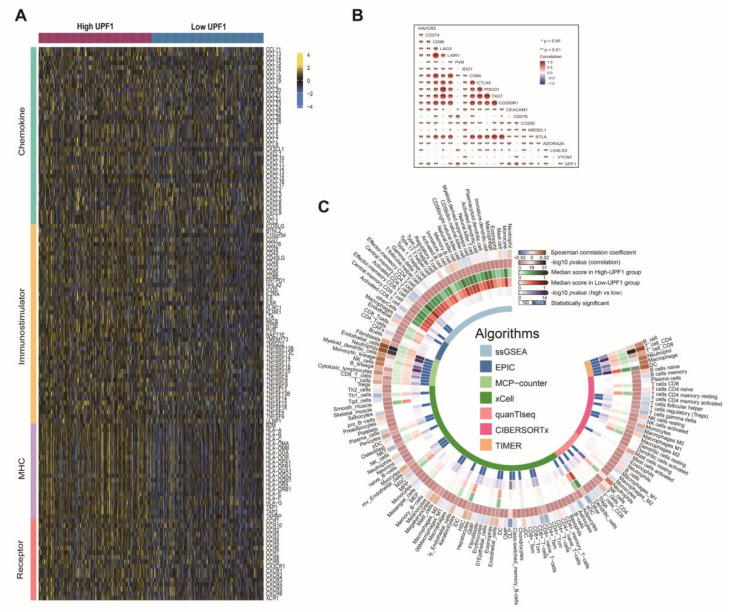
The effects of UPF1 in the ccRCC tumor microenvironment. (**A**) High-UPF1 and low-UPF1 groups of ccRCC express different immunomodulators (chemokines, receptors, MHC, and immunostimulators). (**B**) Identifying the 20 inhibitory immune checkpoints associated with UPF1. (**C**) Immune cells characteristics in ccRCC of high-UPF1 and low-UPF1 groups: The outermost ring represented the correlations between immune cell infiltration levels and UPF1 expression levels; The second outer ring represented the *p* values of correlation coefficients; The third and fourth outer rings represented the median infiltration level of immunocytes in high-UPF1 group and low-UPF1 group; The fifth and sixth outer ring represented the *p* value of differences between immunocyte infiltration levels between high-UPF1 and low-UPF1 groups. The inner ring represented the according algorithms that we employed.

**Figure 4 genes-13-02166-f004:**
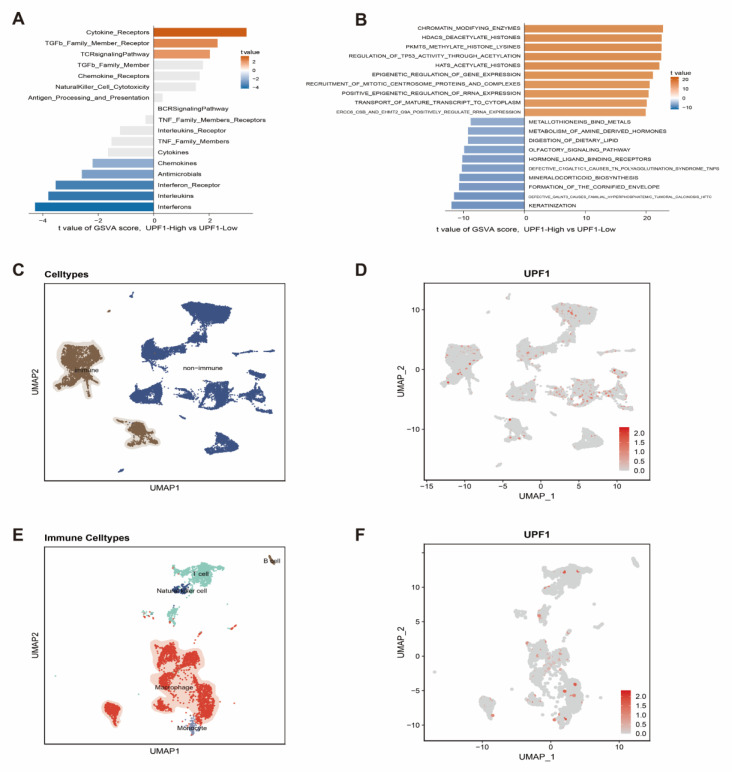
(**A**) Comparing high-UPF1 and low-UPF1 groups in terms of pathway enrichment. (**B**) Comparing high-UPF1 and low-UPF1 groups in terms of Reactome pathway enrichment. (**C–F**) Immune cell type-specific expression of UPF1 in ccRCC.

**Figure 5 genes-13-02166-f005:**
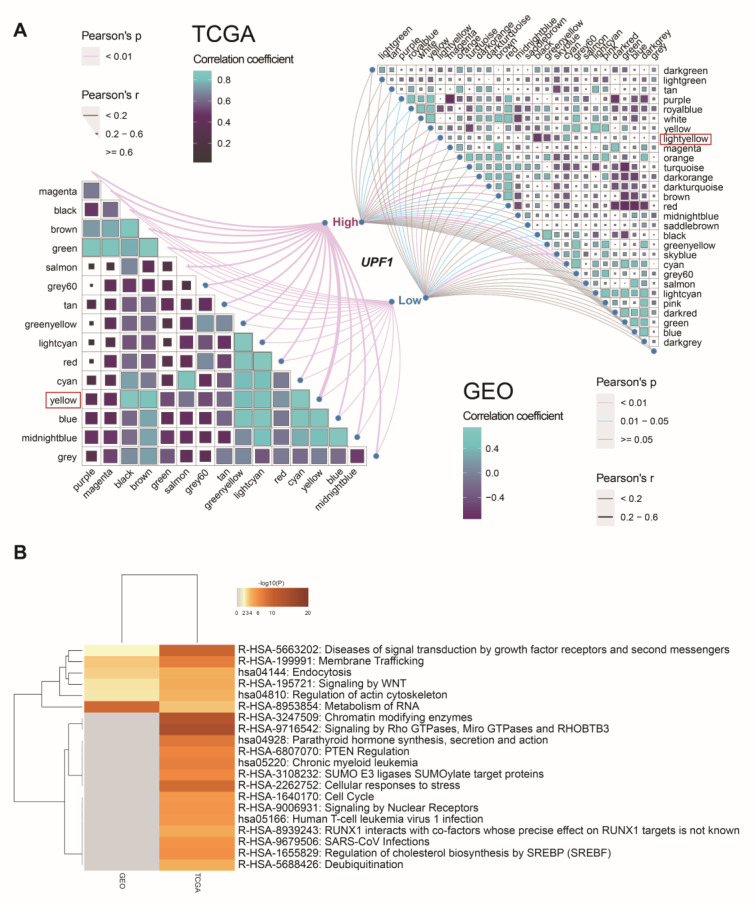
Identification of co-expression genes based on UPF1 expression status. (**A**) Weighted co-expression networks based on UPF1 expression-based groups (the right part: GEO database; the left part: TCGA database). (**B**) KEGG and Reactome pathway enrichment analyses between key modules in TCGA and GEO cohorts.

**Figure 6 genes-13-02166-f006:**
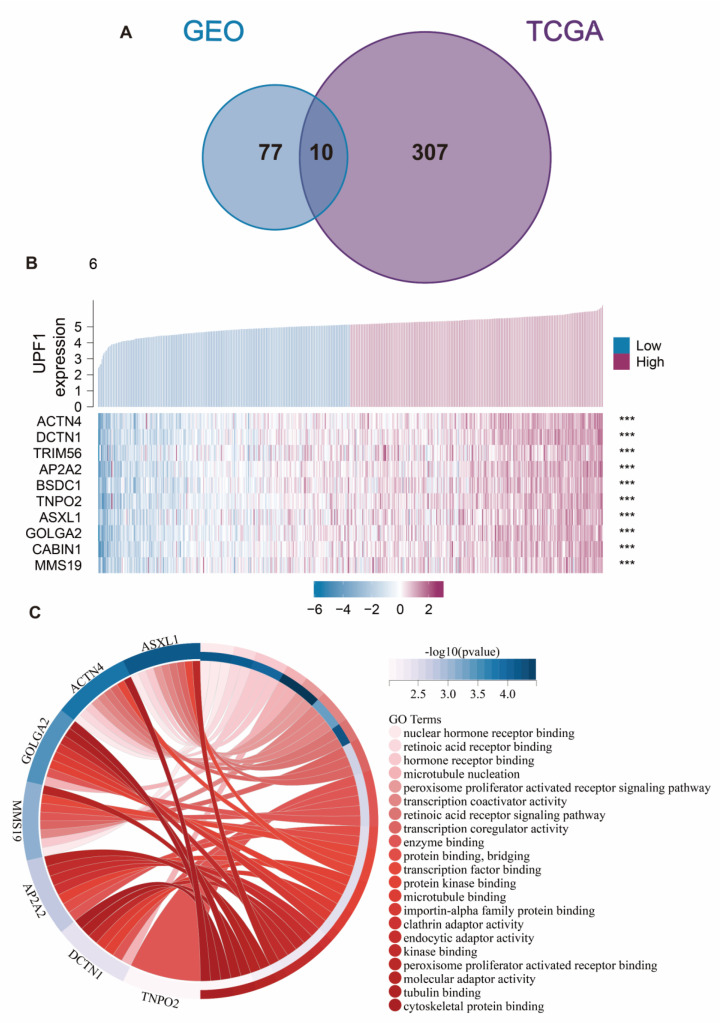
(**A**) Venn diagram shows the intersection of module genes in key modules. (**B**) Heatmap of key co-expression genes expression levels and UPF1 expression levels, and the correlation between them were exhibited, *** *p* < 0.001. (**C**) Chord diagram shows the GO terms that these key co-expression genes were enriched in.

**Figure 7 genes-13-02166-f007:**
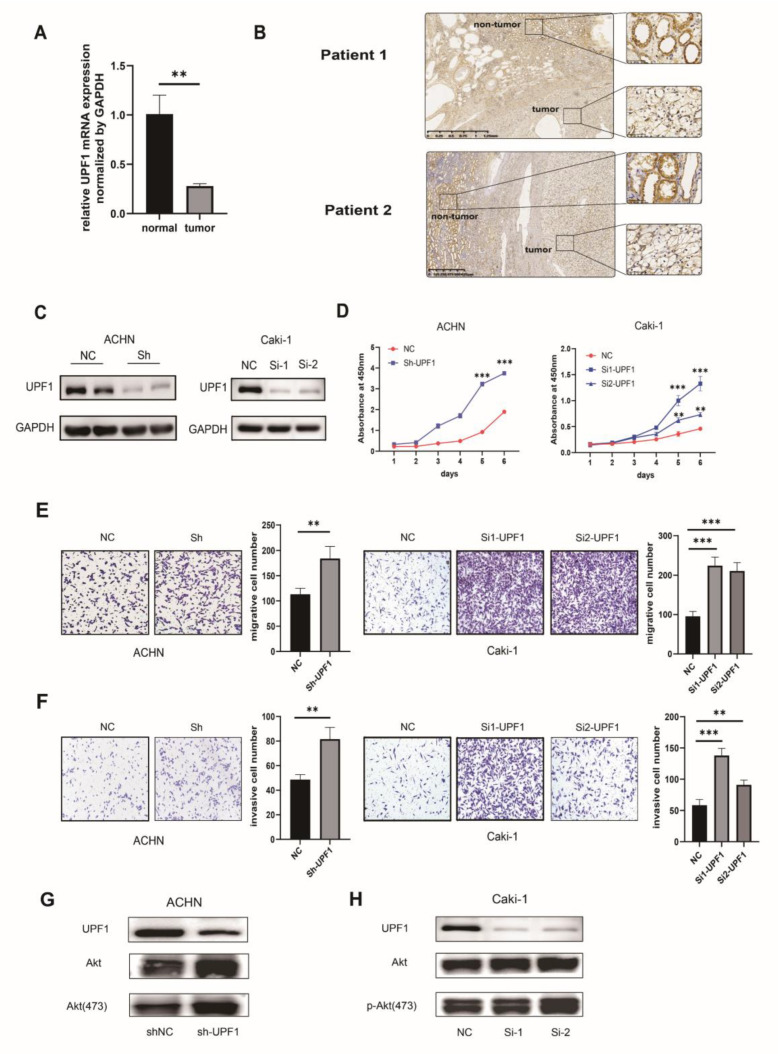
Experimental validation of the role of UPF1 in ccRCC cells. (**A**) qRT-PCR results of UPF1 expression in normal kidney and ccRCC tissues. (**B**) UPF1 protein was strong positive in adjacent nontumor tissues and weak in ccRCC tissues. (**C**) Western blots confirmed the changes in UPF1 expression. (**D**) The proliferation ability of ACHN and Caki-1 cells was measured by the CCK8 assay after transfecting UPF1 shRNA. The migration (**E**) and invasion (**F**) of ACHN and Caki-1 cells were performed in the control group and the si-UPF1 group. (**G**,**H**) Knockdown of UPF1 active the AKT signal pathway in ACHN and Caki-1 cell lines. ** *p* < 0.01; *** *p* < 0.001.

**Figure 8 genes-13-02166-f008:**
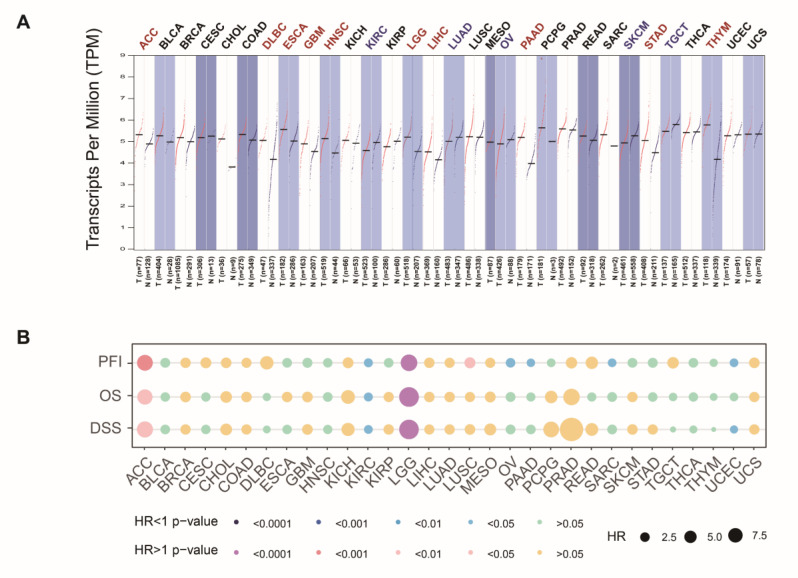
UPF1 pan-cancer analysis (**A**) Expression level of UPF1 in 31 cancer types. (**B**) Disease specific survival (DSS), overall survival (OS), and progression-free interval (PFI) analysis of 31 cancer types based on UPF1 expression.

## Data Availability

We acknowledge TCGA and GEO database for providing their platforms and contributors for uploading their meaningful datasets.

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
