# Peer review of "Identification and Validation of UPF1 as a Novel Prognostic Biomarker in Renal Clear Cell Carcinoma"

_genes, 2022, doi:10.3390/genes13112166_

Round 1

Reviewer 1 Report

The manuscript requires a serious revision.

1. the plots are nice but the font size is way too small and thus the figures will be barely readable/recognizable in the published version.

2. the abstract and conclusions are written in a rather defocused style. the authors have to formulate the problem, what they did, and how did they do it very clearly. these two sections are often the only parts of the text the average reader will read so they must be written crystal-clear and precise.

3. in the discussion section the discussion of and the comparison with other/similar studies in the literature is missing and should be added in the revised version.

4. it is important to emphasize in the discussion how the results of the current study help to unveil the mechanisms of proliferation and principles of functioning of different types of cancer. this should be presented in a rather nonspecific way, accessible to a broad audience of readers.

Author Response

Dear Reviewer:

On behalf of my co-authors, we thank you very much for giving us an opportunity to revise our manuscript. we appreciate you very much for your positive and constructive comments on our manuscript entitled “Identification and validation of UPF1 as a novel prognostic biomarker in renal clear cell carcinoma” (Manuscript ID: genes-2014059).

We have studied your comments carefully and tried our best to revise our manuscript according to the comments. The following are the responses to your comments on an item-by-item basis, and the changes made within the article of this revised manuscript are colored by red.

Responds to the comments:

Main comments: The manuscript requires a serious revision.

Comment 1: the plots are nice but the font size is way too small and thus the figures will be barely readable/recognizable in the published version.

Response: We are grateful to you for pointing out this problem. We have uploaded high-definition images with bigger font size.

Comment 2: the abstract and conclusions are written in a rather defocused style. the authors have to formulate the problem, what they did, and how did they do it very clearly. these two sections are often the only parts of the text the average reader will read so they must be written crystal-clear and precise.

Response: Thank you very much for these valuable suggestions. We have rewritten the abstract and conclusion to ensure that average readers can understand our research clearly and accurately.

Comment 3: in the discussion section the discussion of and the comparison with other/similar studies in the literature is missing and should be added in the revised version.

Response: Thank you very much for your constructive advice. In the discussion part of the revised edition, we have added the review of other research about UPF1 in cancers (Page 18, lines 403-416). As follows:

Finally, Pan-cancer analysis showed that low expression of UPF1 was associated with poor prognosis in multiple cancers. In previous studies, UPF1 was found to be significantly down regulated in pancreatic cancer [1], hepatocellular carcinoma (HCC) [2], gastric cancer (GC) [3], thyroid cancer (TC) [4] and glioma [5], and was verified in most cell lines of these cancers. However, Ha and Bokhari A found that UPF1 was significantly upregulated in LADC [6] and CRC (in MSI) [7]. We propose there might be two main reasons for the inconsistent expression trend of UPF1 in different cancers. Firstly, it was reported that oxidative stress and endoplasmic reticulum (ER) stress could sup-press NMD through the phospho-eIF2α/ATF4 pathway [8]. This might explain the low expression of UPF1 in renal cancer, which was features enhanced oxidative stress. Secondly, tumors with high tumor mutation burdens (TMB) tend to generate a high number of PTC-mRNAs, which will trigger NMD and account for increased expression levels of NMD factors [8]. These findings indicate that the expression of UPF1 is in-consistent in different cancers. However, the mechanism of UPF1 in these cancers needs further study.

  1. Liu, C., et al., The UPF1 RNA surveillance gene is commonly mutated in pancreatic adenosquamous carcinoma. Nat Med, 2014. 20(6): p. 596-8.
  2. Chang, L., et al., The human RNA surveillance factor UPF1 regulates tumorigenesis by targeting Smad7 in hepatocellular carcinoma. J Exp Clin Cancer Res, 2016. 35: p. 8.
  3. Li, L., et al., The Human RNA Surveillance Factor UPF1 Modulates Gastric Cancer Progression by Targeting Long Non-Coding RNA MALAT1. Cell Physiol Biochem, 2017. 42(6): p. 2194-2206.
  4. Zhong, Z.B., et al., Knockdown of long noncoding RNA DLX6-AS1 inhibits migration and invasion of thyroid cancer cells by upregulating UPF1. Eur Rev Med Pharmacol Sci, 2020. 24(16): p. 8246.
  5. Lv, Z.H., Z.Y. Wang, and Z.Y. Li, LncRNA PVT1 aggravates the progression of glioma via downregulating UPF1. Eur Rev Med Pharmacol Sci, 2019. 23(20): p. 8956-8963.
  6. Han, S., et al., LncRNA ZFPM2-AS1 promotes lung adenocarcinoma progression by interacting with UPF1 to destabilize ZFPM2. Mol Oncol, 2020. 14(5): p. 1074-1088.
  7. Bokhari, A., et al., Targeting nonsense-mediated mRNA decay in colorectal cancers with microsatellite instability. Oncogenesis, 2018. 7(9): p. 70.
  8. Usuki, F., A. Yamashita, and M. Fujimura, Environmental stresses suppress nonsense-mediated mRNA decay (NMD) and affect cells by stabilizing NMD-targeted gene expression. Sci Rep, 2019. 9(1): p. 1279.

Comment 4: it is important to emphasize in the discussion how the results of the current study help to unveil the mechanisms of proliferation and principles of functioning of different types of cancer. this should be presented in a rather nonspecific way, accessible to a broad audience of readers.

Response: Thank you to point this out, which will significantly improve the quality of our study. In the discussion section of the revised version, we have emphasized the important role of the current study to help to unveil the mechanisms of proliferation and principles of functioning of ccRCC and other types of cancer (Page 17, lines 394-402). As follows:

The role of endogenous UPF1 was explored in ccRCC cells. UPF1 was silenced in ACHN and Caki-1 cells by using specific siRNA against UPF1. A CCK8 incorporation assay and Transwell assays confirmed that downregulation of UPF1 promoted the pro-liferation, migration and invasion of ccRCC cells. Knockdown of UPF1 in normal renal cell line can also promote proliferation. To explore the underlying mechanism of the tumor suppressive roles of UPF1 in ccRCC, we detected the phosphorylated AKT through Western blotting in UPF1-silenced ACHN and Caki-1 cells, and the results suggested that the AKT signaling might be activated after UPF1 is knocked out. How-ever, further experiments need to be conducted to unveil the molecular mechanisms of UPF1 in ccRCC.

We appreciate for your warm work earnestly,

Kind regards,

Reviewer 2 Report

The manuscript by Shijuan Mai and colleagues describes the impact of low UPF1 expression on the progression of clear cell renal carcinoma (ccRCC). The Authors properly used bioinformatic tools, as well as in vitro experiments, to provide evidence of their thesis. The paper is well organized, contains powerful pictures and represents a valuable contribution to cancer research field. More detailed comments are reported below.

- Please check English language throughout the manuscript.

- Figure 1 should be revised, since it is a collage of the other figures within this manuscript. Some captions are too small and are not properly readable (e.g. co-expression gene analysis).

- Figures 3C, 3D and 3E contain unreadable captions. Please, split the figures or revise them. The same applies to figure 4E.

- Lines 216-217: please correct *p<0.05, **p<0.01 and ***p<0.001.

Author Response

Dear Reviewer:

On behalf of my co-authors, we thank you very much for giving us an opportunity to revise our manuscript. we appreciate you very much for your positive and constructive comments on our manuscript entitled “Identification and validation of UPF1 as a novel prognostic biomarker in renal clear cell carcinoma” (Manuscript ID: genes-2014059).

We have studied your comments carefully and tried our best to revise our manuscript according to the comments. The following are the responses to your comments on an item-by-item basis, and the changes made within the article of this revised manuscript are colored by red.

Responds to the comments:

Main comments:

The manuscript by Shijuan Mai and colleagues describes the impact of low UPF1 expression on the progression of clear cell renal carcinoma (ccRCC). The Authors properly used bioinformatic tools, as well as in vitro experiments, to provide evidence of their thesis. The paper is well organized, contains powerful pictures and represents a valuable contribution to cancer research field. More detailed comments are reported below.

Comment 1: Please check English language throughout the manuscript.

Response: Considering your suggestion, we have carefully corrected the typing errors and English grammatical mistakes in our manuscript, and invited the MDPI office to further refine and polish the English language of our manuscript (ID: english-edited-53718).

Comment 2: Figure 1 should be revised, since it is a collage of the other figures within this manuscript. Some captions are too small and are not properly readable (e.g. co-expression gene analysis).

Response: Thank you for pointing out this problem. We have uploaded high-definition images with bigger font size (Page 3, revised Figure 1).

Comment 3: Figures 3C, 3D and 3E contain unreadable captions. Please, split the figures or revise them. The same applies to figure 4E.

Response: We are grateful to you for pointing out this problem. We have splited original Figure 3A-I to revised Figure 3A-C and revised Figure 4A-D (Page 9-10),and uploaded the revised images of with bigger font size.

Comment 4: Lines 216-217: please correct *p<0.05, **p<0.01 and ***p<0.001.

Response: We are very sorry for our mistake. And we have made correction in the Page 8, lines 245-246.

We appreciate for your warm work earnestly,

Kind regards,

Reviewer 3 Report

The authors used bioinformatic methods to identify that UPF1 has lower expression  in ccRCC. In addition, the authors used knockdown experiments to prove that knocking down UPF1 lead to ACHN and Caki-1 cell proliferation increase and invasion increase. Their finding suggested that low expression of UPF1 is a biomarker in ccRCC.

The authors should address following questions to strengthen the manuscript.

1.  The UPF1 expression in different cancer type showed different trend. For example, the UPF1 has high expression in CRC but low expression in HCC. What would be the reason that UPF1 showed different function in different cancer type? The authors should explain in the manuscript.

2.  In Figure5, the authors knockdown UPF1 in ACHN and Caki-1 cells and the cells showed increased  proliferation. The authors should include a normal cell line and knockdown UPF1 in normal cell lines, what would be the proliferation curve?

3.  What is the mechanism of UPF1 knockdown induced cell proliferation decrease and invasion decrease? The author should check the UPF1 downstream genes mRNA and protein level to find which pathway is involved here.

Author Response

Dear Reviewer:

On behalf of my co-authors, we thank you very much for giving us an opportunity to revise our manuscript. we appreciate you very much for your positive and constructive comments on our manuscript entitled “Identification and validation of UPF1 as a novel prognostic biomarker in renal clear cell carcinoma” (Manuscript ID: genes-2014059).

We have studied your comments carefully and tried our best to revise our manuscript according to the comments. The following are the responses to your comments on an item-by-item basis, and the changes made within the article of this revised manuscript are colored by red.

Responds to the comments:

Main comments:

The authors used bioinformatic methods to identify that UPF1 has lower expression in ccRCC. In addition, the authors used knockdown experiments to prove that knocking down UPF1 lead to ACHN and Caki-1 cell proliferation increase and invasion increase. Their finding suggested that low expression of UPF1 is a biomarker in ccRCC. The authors should address following questions to strengthen the manuscript.

Comment 1: The UPF1 expression in different cancer type showed different trend. For example, the UPF1 has high expression in CRC but low expression in HCC. What would be the reason that UPF1 showed different function in different cancer type? The authors should explain in the manuscript.

Response: Thank you very much for pointing out this issue. We explained it in the manuscript in Page 18, lines 408-416, as follows:

We propose there might be two main reasons for the inconsistent expression trend of UPF1 in different cancers. Firstly, it was reported that oxidative stress and endoplasmic reticulum (ER) stress could suppress NMD through the phospho-eIF2α/ATF4 pathway [1]. This might explain the low expression of UPF1 in renal cancer, which was features enhanced oxidative stress. Secondly, tumors with high tumor mutation burdens (TMB) tend to generate a high number of PTC-mRNAs, which will trigger NMD and account for increased expression levels of NMD factors [2]. These findings indicate that the expression of UPF1 is inconsistent in different cancers.

  1. Usuki, F., A. Yamashita, and M. Fujimura, Environmental stresses suppress nonsense-mediated mRNA decay (NMD) and affect cells by stabilizing NMD-targeted gene expression. Sci Rep, 2019. 9(1): p. 1279.
  2. Bokhari, A., et al., Targeting nonsense-mediated mRNA decay in colorectal cancers with microsatellite instability. Oncogenesis, 2018. 7(9): p. 70.

Comment 2: In Figure5, the authors knockdown UPF1 in ACHN and Caki-1 cells and the cells showed increased proliferation. The authors should include a normal cell line and knockdown UPF1 in normal cell lines, what would be the proliferation curve?

Response: We appreciate this kind comment. In order to explore the mechanism of UPF1 deletion promoting the proliferation of renal cell carcinoma cell lines, we knocked out UPF1 of normal renal cell lines. The CCK8 increment curve suggested that the knockdown of UPF1 would also promote the proliferation of normal renal cell lines (Page 14, lines 325-327, Figure S5).

Comment 3: What is the mechanism of UPF1 knockdown induced cell proliferation decrease and invasion decrease? The author should check the UPF1 downstream genes mRNA and protein level to find which pathway is involved here.

Response: Thank you for this valuable feedback. The enrichment analysis of UPF1 co-expression genes indicated the involvement of PTEN regulation. Since PTEN is the key upstream regulator of AKT signaling pathway, we conducted a western blot assay and found that the phosphorylated form of AKT obviously increased, suggesting the activation of AKT signal pathway after UPF1 knockdown in ACHN and Caki-1 cells (Page 14, lines 329-333, Figure 6G, H).

We appreciate for your warm work earnestly,

Kind regards,

Round 2

Reviewer 1 Report

satisfactory revision

Author Response

Thanks.